# Human-Robot Joint Misalignment, Physical Interaction, and Gait Kinematic Assessment in Ankle-Foot Orthoses

**DOI:** 10.3390/s24010246

**Published:** 2023-12-31

**Authors:** Ricardo Luís Andrade, Joana Figueiredo, Pedro Fonseca, João P. Vilas-Boas, Miguel T. Silva, Cristina P. Santos

**Affiliations:** 1Center for MicroElectroMechanical Systems (CMEMS), University of Minho, 4800-058 Guimarães, Portugal; 2LABBELS—Associate Laboratory, 4710-057 Braga/4800-058 Guimarães, Portugal; 3Porto Biomechanics Laboratory (LABIOMEP), University of Porto, 4200-450 Porto, Portugal; pedro.labiomep@fade.up.pt (P.F.); jpvb@fade.up.pt (J.P.V.-B.); 4Centre of Research, Education, Innovation and Intervention in Sport (CIFI2D), Faculty of Sport, University of Porto, 4200-450 Porto, Portugal; 5IDMEC, Instituto Superior Técnico, Universidade de Lisboa, Av. Rovisco Pais 1, 1049-001 Lisbon, Portugal; miguelsilva@tecnico.ulisboa.pt

**Keywords:** exoskeletons and ankle-foot orthoses, human-exoskeleton misalignment, human-robot physical interaction, gait kinematics, rehabilitation robotics

## Abstract

Lower limb exoskeletons and orthoses have been increasingly used to assist the user during gait rehabilitation through torque transmission and motor stability. However, the physical human-robot interface (HRi) has not been properly addressed. Current orthoses lead to spurious forces at the HRi that cause adverse effects and high abandonment rates. This study aims to assess and compare, in a holistic approach, human-robot joint misalignment and gait kinematics in three fixation designs of ankle-foot orthoses (AFOs). These are AFOs with a frontal shin guard (F-AFO), lateral shin guard (L-AFO), and the ankle modulus of the H2 exoskeleton (H2-AFO). An experimental protocol was implemented to assess misalignment, fixation displacement, pressure interactions, user-perceived comfort, and gait kinematics during walking with the three AFOs. The F-AFO showed reduced vertical misalignment (peak of 1.37 ± 0.90 cm, *p*-value < 0.05), interactions (median pressures of 0.39–3.12 kPa), and higher user-perceived comfort (*p*-value < 0.05) when compared to H2-AFO (peak misalignment of 2.95 ± 0.64 and pressures ranging from 3.19 to 19.78 kPa). F-AFO also improves the L-AFO in pressure (median pressures ranging from 8.64 to 10.83 kPa) and comfort (*p*-value < 0.05). All AFOs significantly modified hip joint angle regarding control gait (*p*-value < 0.01), while the H2-AFO also affected knee joint angle (*p*-value < 0.01) and gait spatiotemporal parameters (*p*-value < 0.05). Overall, findings indicate that an AFO with a frontal shin guard and a sports shoe is effective at reducing misalignment and pressure at the HRI, increasing comfort with slight changes in gait kinematics.

## 1. Introduction

Diseases like cerebral palsy, stroke, or ataxia commonly result in a vast array of symptoms for the individual, such as asymmetrical/abnormal gait patterns, loss of balance, and muscle spasticity. Generally, these patients require gait rehabilitation [1,2,3], where lower limb robotic devices like exoskeletons and ankle-foot orthoses (AFOs) have gained increased importance in the last decade [1,2]. Exoskeletons can strengthen muscle action through torque transmission, thereby reproducing functional gait patterns [3] and providing support for lower limbs in different conditions, such as standing in an upright position [3,4]. Moreover, AFOs have also been used to assist patients with drop foot [5,6,7], improving ankle joint alignment and walking speed [8]. Furthermore, the active AFOs can actively assist with ankle dorsiflexion [9] and increase foot clearance from the ground [9,10].

Despite the increasing number of exoskeletons and AFOs being developed and the ones commercially available, the interface between humans and robots has not been properly designed [11]. A third of the assistive AFOs are dropped by their users due to usability limitations, such as disturbance of the human gait patterns and soft and musculoskeletal tissue injuries [12,13,14,15]. For anthropomorphic AFOs, defined in [15] as the AFOs “where any hinge corresponds to a degree of freedom (DOF) of the human limb”, a proper alignment of these hinges (or robotic joints) with the biological joints is paramount to minimizing these usability limitations [16]. The misalignment between the robotics and biological joints results in spurious forces and torques at the physical human-robot interface (HRi) [17], leading to discomfort, pain, or long-term injury [12,13,14,15,16,18,19,20,21]. It is needed to develop robotic AFOs with alignment solutions to correct the misalignment.

To properly develop an alignment solution, we first need to effectively assess the magnitude of misalignment and its effects on the AFOs. The misalignment assessment can be done through three avenues. First, misalignment can be assessed directly through motion capture systems. In [21], the misalignment between the user’s knee joint and an active knee exoskeleton was measured by infrared-based motion capture. To do this, both joints were mapped using infrared markers, and their position in space was calculated. The difference between the positions of the two joints in the sagittal plane was computed as a measure of joint misalignment. In [22], a similar process was employed on a dummy limb that mimicked the knee joint by computing rotational misalignment. The misalignment can lead to shear and pressure interactions [11,13], which can be measured as a second avenue to assess misalignment. Shear interactions can be measured indirectly through infrared-based motion capture by assessing the relative displacement between the AFO’s fixations (e.g., cuffs and straps) and the user’s soft tissue [23,24,25]. Studies have used this method to measure deformations on lower limb orthoses [25] and to detect cuff slippage in a lower limb exoskeleton [24]. Within these studies, the relative displacement between two markers, one in the user and one in the strap/cuff is computed. Regarding the third avenue, several studies have captured pressure peaks and the pressure distribution at the HRi for safety validation [13]. This has been done in [26,27,28] by using force-sensitive resistors (FSRs) to assess peak pressure values and compare them with standard safety values. A systematic review of HRi measurement in lower limb exoskeletons, orthoses, and prostheses found that most studies had measured pressure interactions through FSR-based technology [29].

To the best of our knowledge, there are three different fixation designs at the shank level for AFOs, namely straps [30], a frontal shin-guard [31], and a lateral shin-guard [32]. A literature review of 26 AFOs [33] verified that 18 studies used straps with no lateral or frontal support from a shin guard, and the remaining 8 AFOs included lateral and/or frontal support for load distribution. However, no study has presented a holistic comparison between possible fixation designs for the physical HRi prior to making the design choices. There is still no knowledge regarding which fixation design leads to the least adverse effects on the user (such as misalignments, pressure peaks, and discomfort).

Thus, the main goal of this work is to perform a holistic assessment to quantify the human-robot joint misalignment and gait kinematic effects of three AFOs and, based on this assessment, to compare the three fixation designs of these AFOs. One of the AFOs is a commercially available AFO (H2-AFO) that interfaces with the user’s shank through straps [34], while the other AFOs were developed in-house with a lateral and frontal shin guard. This study hypothesizes that the three prototypes have significant differences regarding misalignment and gait kinematic effects. The assessment was performed during level-ground walking at different speeds through three sources of data: (1) motion capture data (assess the human-AFO joint misalignment, fixation displacement, and gait kinematics); (2) pressure at the Hri measured by FSR (assess the human-robot interactions); and (3) a comfort and satisfaction questionnaire (assess the user’s perceived comfort of the prototypes).

To the best of our knowledge, this work is the first to assess and correlate different objective measures of human-AFO joint misalignment, such as pressure at HRi, fixation displacement, and joint misalignment, together with user-perceived comfort, to get a holistic perception of misalignment effects on humans. Furthermore, it is the first study that combines these measures with human gait kinematics to present a holistic comparative analysis of the effects of three different fixation designs on human walking. This analysis aims to answer the following question: which fixation design (straps, frontal shin guard, and lateral shin guard) leads to decreased misalignment, related interactions, gait disturbance, and higher user satisfaction? Finally, this analysis may guide future studies on AFO fixation design, human-robot misalignment measurement, and designing alignment solutions by making a direct assessment of the measures needed to be compensated.

## 2. AFOs Description

Our work included three AFOs, two team-developed models, and the ankle module of the H2 exoskeleton (H2-AFO). They differ in the fixation mechanisms of the shank and foot segments. The two prototypes developed in-house were denominated SmartOs Frontal (F-AFO, presented in Figure 1a) and SmartOs Lateral (L-AFO, Figure 1b) according to the frontal and lateral locations of their shin guards (1 in Figure 1a,b), respectively [35]. The frontal and lateral shin guards have a thickness of 2 mm and are made of aluminum (AL5754), which is coated with neoprene. For attaching the shin guard to the user’s shank, the F-AFO and L-AFO use Velcro (2 in Figure 1a) and straps (2 in Figure 1b), respectively. Further, both shin guards are attached to an aluminum structure (AL5754) that allows for manual alignment to fit users with heights between 1.70 and 1.90 m and a maximum body mass of 100 kg. Both designs included a commercially available sports shoe (European size 42) at the foot segment (Wock Breelite, Wock Shoes, 3 in Figure 1a,b) to improve comfort and ergonomics. The actuator and electronics have been theorized but not yet implemented in these prototypes. These models have a stainless-steel structure (6061-T6) at the ankle joint level (4 in Figure 1a,b) that mimics both the volume and mass of these electronics, guaranteeing that the structure’s inertia is approximately the same as if the whole apparatus were present. The F-AFO weighs 2.18 kg, and the L-AFO weighs 1.90 kg.

The H2 exoskeleton (Technaid S.L., Spain) is a robotic exoskeleton developed to assist impaired human walking using six actuated joints, namely both hip, knee, and ankle joints [17]. The exoskeleton is fully modular, allowing the user to wear any of the joints or any combination of joints. The ankle modulus (H2-AFO, illustrated in Figure 1c) used for this work was designed for adults with a height ranging from 1.50 to 1.95 m and a maximum body mass of 100 kg. Fixation to the user is ensured at three locations: two adjustable straps at the shank (2 in Figure 1c) and a specific outsole platform (European size 42, 3 in Figure 1c) that connects to the foot. The fixations at the shank level are made through adjustable Velcro straps with foam pads to minimize pressure. The mechanical structure is made of stainless steel and 7005 aluminum. The assistance of the H2-AFO can be controlled by different control strategies, including the zero-impedance controller. This control algorithm allows the AFO to follow the user’s gait speed and motion intention, which are detected by strain gauge sensors embedded in the shank structure, acting with passive behavior. This AFO weighed 2.1 kg. Table 1 details the characteristics of each AFO, including materials and mass.

## 3. Materials and Methods

In this section, we detail the experimental protocol, instrumentation, and data analysis process for assessing the misalignment, pressure interactions, and user comfort and satisfaction during overground walking while wearing the AFOs described in Section 2.

### 3.1. Participants

Misalignment and interactions were assessed on 10 young male healthy subjects (13 ± 4.0 cm, 81.1 ± 10.1 kg, and 25.8 ± 4.4 years old). All participants were healthy without reporting any known locomotion or balance impairment, and they had not suffered any musculoskeletal injury in the previous six months. We selected male participants given the height restrictions of the available L-AFO and F-AFO since it was not possible to have an equal distribution of female and male participants within the height range of 1.70–1.90 m. All participants were informed of the study’s objectives and methodology and received an informed consent form, which they read and signed. The study was approved by the University of Minho Research in Life and Health Sciences Ethics Committee, with the protocol number CEICVS 006/2020. Table 2 presents the average distance between each strap of each AFO and the popliteal fossa in centimeters and as a percentage of shank length (in parentheses).

### 3.2. Instrumentation and Data Collection

A motion capture system with 12 cameras (Oqus, Qualisys—Motion-Capture System, Göteborg, Sweden) was used to measure, at 100 Hz, the kinematics of human lower limbs (marker set in Figure 2) and AFOs (marker sets illustrated in Figure 3, Figure 4 and Figure 5). Overall, we used 30 retro-reflective markers on the human body and 9 markers for each AFO. The latter set was chosen to create two (F-AFO and L-AFO) and three (H2-AFO) rigid bodies for further kinematic analysis. All markers were placed by the same operator using the anatomical standards described in [36].

Pressures at the interfaces between the user and the AFO were measured by a system of 8 circular FSRs (FSR 400 model, Interlink Electronics, Irvine, CA, USA). These sensors change their resistance when force is applied, outputting a voltage. This voltage was received and processed by an Arduino Nano microcontroller board, operating at 100 Hz. Each FSR was calibrated prior to the protocol, using the method described in [37] for pressure steps of 0, 20, 25, 35, 50, 75, 100, and 150 KPa. Results were fitted to an exponential model, as presented in Equation (1), through the curve fitting tool of MATLAB R2020b (The MathWorks, Natick, MA, USA).
(1)y=a·exp(b·x)+c·exp(d·x)
where, *x* represents the digital voltage signal captured by the system and *y* is the resulting pressure, in KPa. Constants *a*, *b*, *c,* and *d* were defined for each FSR individually.

We labeled each FSR sensor from 1 to 8 and fixed it to the calf of AFOs through double-sided tape, as illustrated in Figure 6. We placed the FSRs in anatomical areas where high-pressure values should be avoided due to the increased risk of discomfort and injuries [17,38,39]. The FSR placement on AFOs was done by the same operator throughout the protocol. Once the user donned the AFO, the operator repositioned the FSRs to ensure that they were repeatedly (across subjects) positioned in the same anatomical landmarks as the user’s shank. This is especially important for FSR 3 and FSR 6 of F-AFO, FSR 6 and FSR 8 of L-AFO, and FSR 3 and FSR 8 of H2-AFO, where the intent was to place them at the bony prominence of the tibia and not on the adjacent softer tissue. The motion capture and pressure data were captured in synchrony. Regarding the interactions measured, no direct assessment was made of sheer forces, and motion capture data was used as a surrogate for quantifying these interactions.

The comfort and satisfaction questionnaire included 13 questions, scored on a 4-point Likert scale (Strongly Disagree, Disagree, Agree, and Strongly Agree). The used questionnaire was based on an available standard questionnaire for assistive devices, the adapted Client Satisfaction with Orthotics and Prosthetic Users Survey (CSD-OPUS) [40]. This questionnaire assesses the user’s perception of the device in various dimensions. Importantly, it addresses the presence of abrasion and irritation on the user’s skin. This is commonly the result of friction interactions at the level of the HRi, which are one of the main effects of misalignment [12,13,19,21]. The original CSD-OPUS questionnaire can be found in Appendix B.

Thus, the questionnaire issued in this study used the original questions numbers 1, 4, 5, 6, 7, 10, 11, and 12 from the modified CSD-OPUS. Questions 2 and 6 of the original questionnaire were expanded into questions 2, 3, 8, and 9 of our questionnaire in order to assess differences between the shank and foot moduli. Question 13 was added since there was a need to assess the user’s opinion of the dimensions of the AFO, like size, height, length, or width. Figure 7 presents the questionnaire used in this study.

### 3.3. Experimental Protocol

An operator donned the participants with one of the three AFOs (chosen in random order) on their right leg. The operator was constant for all participants. Participants were asked to wear either sports shorts or skin-tight long tights, as conducted in the clinical practice of rehabilitation. Our experience shows that wearing either of these garments does not lead to confounding results.

The protocol started by recording participant’s anthropometric data, such as body mass, height, shank length, and shank perimeter. After that, the participants were asked to perform three trials of overground walking for 10 m at a self-selected slow speed using the AFO. The participants further repeated these walking trials at a controlled cadence of 70 steps per min (roughly equivalent to 1.6 km/h, the maximum speed allowed by the H2-AFO), with the help of a metronome. Then, the user walked on a treadmill for 6 min at 1.6 m/s (data not analyzed in this work). The protocol ended by doffing the AFO (helped by the same operator) and answering a questionnaire on his/her experience with this AFO. This protocol (illustrated in Figure 8) was repeated for each AFO. It is important to note that during the experiment, the assistance of the H2-AFO was controlled by a zero-impedance control to approximate the behavior of this AFO to passive assistance.

Finally, all participants were asked to perform three overground trials at self-selected speed without any AFO, which will be used as control of gait trials. A video of the protocol for each AFO is provided in Appendix A.

### 3.4. Data Processing and Analysis

For processing the motion capture data, first, the retro-reflective markers were identified by a dedicated operator through the identification method of the Qualisys Track Manager software version 2021.2.6940. The marker’s trajectory was fitted to a polynomial interpolation function. Further processing of the motion capture data was done in Visual 3D version 2020.03.26 (C-Motion, Boyds, MD, USA). All markers’ trajectories were filtered through a 6th-order low-pass Butterworth filter with a cutoff frequency of 6 Hz. Human joint centers were defined as the midpoint between the lateral and medial markers. The human segments were defined proximally by the relevant joint and distally by the lateral and medial markers. Both foot segments were defined by the lateral and medial markers in both instances. Human segment masses were determined as proportions of body mass through anthropometric equations native to the Visual 3D software. The used models to define each segment of the three AFOs and the human body are presented in Appendix C.

(1)
*Misalignment-related measures*


Four different misalignment-related measures were computed from the kinematic data using Visual 3D. The measures of *Misalignment distance* and *Misalignment angle* were used to assess human-AFO joint misalignment. The *Strap displacement* and *Shin guard/strap angular displacement* aimed to indirectly assess interactions at the HRi by measuring fixation displacement (linear and angular) between the strap/shin guard and the user’s shank, as proposed by [25]. We computed the four measures, as follows. Figure 9 further details how these measurements were calculated.

*Misalignment Distance*(My, Mz)—defined as the difference between the human joint’s position and the AFO joints in the sagittal plane. Figure 9a) details the markers used for this calculation. Differences were calculated using the axis shown.*Misalignment angle* (Mα, Mβ, and Mγ, corresponding to rotations x-x, y-y, and z-z)—defined as the angle between the human shank and the “support” segment (segments in green in Figure A2), calculated using the JOINTANGLE native function of Visual 3D. This function requires seven main inputs: Segment (Support); Reference Segment (Right Shank); Normalization (Off); 249 Cardan Sequence (X-Y-Z); and negate each of the axes (false for each). Figure 9a) details the markers used for this calculation. Rotations were calculated using the axis shown.*Strap displacement* (Dx, Dy, and Dz)—defined as the strap’s relative movement to the user’s shank in the local reference frame (i.e., the right shank), calculated by the TARGETPATH function of Visual 3D. This function requires three main inputs: Target (markers labeled with an 8 in Figure 3, Figure 4 and Figure 5); Reference Segment (Right Shank); and Resolutions Coordinate System (Right Shank). Figure 9b) details the marker used for this calculation for the F-AFO, identified as STRAP1.*Shin guard/strap angular displacement* (Dα, Dβ, and Dγ corresponding to rotations x-x, y-y, and z-z, respectively)—defined as the angle between the shin-guard (F-AFO and L-AFO) or the top strap (H2-AFO) segment and the user’s right shank, calculated using the Visual 3D function JOINTANGLE. The same inputs were used as in point 2, with the only difference being the chosen segments. Figure 9b) details the marker used for this calculation for the F-AFO.

We removed the initial offset of the angular misalignment and displacement measures. For the misalignment angle, this is important since the reproducibility of marker position across subjects cannot be guaranteed. As such, this systematic error in marker placement will lead to initial angles that vary between subjects. Displacement measures, by definition, should be offset by their initial value.

The signals obtained for these four measures were segmented into gait cycles and each gait cycle into gait phases (stance and swing phases), normalized to 101 points and averaged for all gait cycles using Visual 3D. This was done separately for each set of conditions (subject, speed, and AFO). A complete description of available functions and commands for the software’s pipeline can be found at [41]. Signals were averaged across all 10 subjects and plotted for the three AFOs to help assess the main differences.

(2)
*Pressure on Human-AFO Interaction*


For the processing of pressure data, first, we computed the right heel strike event using Visual 3D. Then, we segmented, in MATLAB, each FSR signal into gait cycles according to the corresponding right heel strike event. The maximum/peak pressure value of each gait cycle was extracted. To make a direct comparison between AFOs, it was necessary to group FSRs placed in the same anatomical location across AFOs. We created four different groups per AFO, as presented in Table 3. Boxplots comparing the peak pressure values within each group were computed to better assess differences between AFOs.

Literature showed that the pain pressure threshold (PPT, defined as the limit of pressure above which a person feels pain) can be measured through single-point algometry [17], while the pain detection threshold (PDT, defined as the pressure magnitude at which pain occurs and analogous with PPT for circumferential pressure) and the pain pressure threshold (PTT, defined as the pressure magnitude that causes unbearable pain) can be assessed through cuff pressure algometry [42,43]. As such, safety was assessed by comparing (i) the peak pressure of all clusters against PTT values captured through single point algometry [17] and (ii) the peak pressure of the posterior cluster against PDT and PPT values captured through CPA [42,43]. The posterior clusters were chosen for safety assessment since (i) the pressures recorded more closely relate to CPA measures and (ii) they correspond to anatomical locations of lower pressure tolerance, as shown by the PTT values for these locations.

(3)
*Analysis of questionnaire responses*


The participant’s answers to the questionnaire were translated to a numeric score by assigning values 1 (Strongly Disagree), 2 (Disagree), 3 (Agree), and 4 (Strongly Agree) to each answer. The scores of the 8 questions taken from the CSD-OPUS were summed and translated to Rasch measures using the table from [40]. To better assess which questions scored higher between the AFOs, the score of all 13 questions was summed for each participant and AFO. Answers of “Not Applicable” were given a 0. Then, F-AFO’s individual question scores were subtracted from those of the other AFOs and plotted in a bar chart. This was done in Microsoft Excel (Version 2208).

(4)
*Gait Kinematics*


For the kinematic gait analysis, we proceeded as follows: We used native functions of Visual 3D to compute human 3D joint angles. The sagittal joint angles of the six lower limb joints (right and left ankles, knees, and hips) were segmented into gait cycles, normalized to 101 points, and averaged for all gait cycles of each condition (the three AFOs and the control trials) using Visual 3D. We also computed the maximum and minimum joint angles for each subject in MATLAB. The spatiotemporal parameters (gait speed and step length, stance time, and swing time of both limbs) were computed and averaged across all trials for each subject and trial condition (trial using AFO and control trial) using Visual 3D. The step length, stance time, and swing time ratios between the left and right limbs were computed in MATLAB.

(5)
*Statistical Analysis*


Table 4 describes the statistical tests carried out in MATLAB to assess the presence or absence of significant differences among trial conditions (gait speed and orthosis). All statistical tests were preceded by one-sample Kolmogorov-Smirnov tests (null hypothesis that the measures come from a normal distribution, 5% significance level). If the data followed a normal distribution, Barttlet tests (null hypothesis that the measures being tested come from a normal distribution with equal variance, 5% significance level) were performed to assess for equal variance between conditions. If no significant differences between variances were found, we used ANOVA tests to assess statistically significant differences between conditions. For non-parametric data, Friedman and Wilcoxon signed-rank tests were conducted to assess for effects between conditions. When significant differences were found, post-hoc Tukey’s HSD tests were conducted both for parametric and non-parametric data.

## 4. Results

### 4.1. Misalignment and Displacement Measures

This section presents the results of *misalignment distance* (Figure 9), *misalignment angle* (Figure 10), *strap displacement* (Figure 11), and *shin-guard/strap angular displacement* (Figure 12), averaged for 10 subjects and plotted for each AFO. Results of the statistical tests described in Table 4 showed no significant differences between gait speeds for the same AFO (*p* < 0.05) for all measures, both during stance and swing phases. As such, the following assessment was done with data from the self-selected speed trials since it more closely resembles a normal gait pattern.

The initial misalignment distances were −1.11 cm, −0.90 cm, and −0.48 cm horizontally and −1.75 cm, −1.82 cm, and −2.76 cm vertically, for F-AFO, L-AFO, and H2-AFO, respectively. The results indicate that there are differences in misalignment distance between AFOs, most notably regarding *vertical misalignment* (*Mz)*. For this measure, H2 is the AFO that presents the highest misalignment at the beginning of the trial. Figure 10 also shows that the misalignment distances (*Mz* and *My*) vary across gaits. Further, *horizontal* and *vertical misalignment* for all three AFOs seems to follow the same pattern throughout the gait cycle.

Regarding the misalignment angles Mα and Mβ (Figure 11), the three AFOs present a similar variation during the stance phase but with symmetrical values (approximately, −1° to 2° for the H2-AFO and −2° to 1° for the F/L-AFO). For Mγ, the three AFOs have approximately the same values.

Regarding linear displacement values (Figure 12), the three AFOs follow generally the same pattern, with values very close in magnitude (<1 cm apart for all measures, except during the swing phase of vertical displacement). Angular displacements (Figure 13) show the same results as angular misalignment. These two measures indirectly represent interactions between the fixations and the user’s soft tissues.

Overall, all four measures indicate no differences between the two models developed by our team. Further tests were done to assess statistically significant differences between AFOs for each of the measures. These were done separately for the stance and swing phases. Table 5 and Table 6 show the mean and standard deviation (during stance and swing, respectively) for joints where statistically significant differences were found between orthosis (*p*-value < 0.05). Furthermore, Table 3 and Table 4 also show the results of the post hoc tests when there were statistically significant differences between the pair of orthoses being tested (*p*-value < 0.05).

Regarding *linear misalignment* measures, statistical test results show significant differences between the H2-AFO and the in-house AFOs for both stance and swing phases, with the H2-AFO having significantly higher misalignments. Furthermore, *linear displacement* measures were significantly different between the H2-AFO and the other two AFOs for Dy and Dz during the stance phase and for three axes during the swing. Finally, the same significant differences in *displacement angles* are also present for Dα.

### 4.2. Pressure on Human-AFO Interface

Here are presented the results regarding the peak pressure measures on the Human-AFO interface to assess the Human-AFO interactions. We evaluated if the gait speed has a significant impact on the interactions at the HRi. The results of Wilcoxon signed-rank tests, presented in Table 7, indicate that for most conditions, there are no statistically significant differences in peak pressure values between both speeds (self-selected and controlled). This suggests that the gait speed does not significantly alter pressure interactions at the HRi. As such, for the following tests, only results for the self-selected speed were considered since this condition most closely resembles a healthy gait pattern.

Figure 14 presents the boxplots obtained for each FSR group. By analyzing Figure 14, we verify that the F-AFO had lower pressure for the four anatomical locations, while the L-AFO and H2-AFO have somewhat similar results for the peak pressure values in the two posterior groups (i.e., posterior shank). The higher differences in peak pressure were observed in the anterior groups, mainly due to the fitness of the frontal shin guard of F-AFO when compared with the other two fixation mechanisms.

Table 8 presents a comparison of the peak pressure values with safety thresholds from the literature [17,42,43]. The pressure values measured for the F-AFO were below the PDT for healthy subjects and patients, except for the posterior proximal group, while for the L-AFO, only the posterior proximal group had pressures below the PDT for healthy individuals. Further, the posterior groups of the H2-AFO show values close to detection thresholds.

### 4.3. Questionnaire on User’s Satisfaction

Comfort was assessed by each participant for each AFO after completing the same walking protocol. Figure 15 presents the sum of scores of the questions from the CSD-OPUS (blue bars), the corresponding Rasch Measures (grey bars), and the sum of scores of all 13 questions (orange bars), with horizontal lines representing the maximum value of each score/measure. Questionnaire scores per participant and AFO, including average scores and pairwise comparisons, are presented in Appendix A.

The Friedman test applied to the Rasch measures indicates a *p*-value of 0.0031 (*p* < 0.05), indicating that there are statistically significant differences in the user’s perception regarding the three AFOs. The results of the post-hoc Tukey test report a higher user satisfaction for the F-AFO than other AFOs (*p*-values of 0.016 and 0.006 < 0.05 for the H2 and L-AFO, respectively), with no clear difference between H2-AFO and L-AFO (*p*-value of 0.9368).

Figure 16 presents the subtraction of F-AFO’s scores from the scores assigned to H2-AFO and L-AFO for each question. It allows us to inspect which questions in F-AFO had higher scores than the other AFOs. Overall, the scores were closer between F-AFO and L-AFO (average difference of 0.45) than between F-AFO and H2-AFO (average difference of 0.58).

The main differences are clear for each AFO. F-AFO has higher scores than H2-AFO mainly in relation to the foot module (questions 2 and 8), shank module (question 3), and overall comfort and fitness (question 4), with scores more than 1 point apart on the Likert scale. The participants reported higher satisfaction scores for F-AFO than L-AFO regarding the pain during the AFO’s use (question 6), the ability to don the AFO (question 11), the fitness of the AFO (question 10), and the presence of abrasions and irritation (question 1).

### 4.4. Gait Kinematics

The ANOVA statistical tests showed no significant differences between maximum and minimum joint angle values across both speeds (1% significance level, in line with similar studies that use a significance level lower than 5% [44,45]). As such, further analysis will focus on self-selected speed trials since control trials were done at this speed and it most closely resembles a healthy gait pattern. Figure 16 presents an overview of results regarding AFOs’ effects on lower limb joint angles. It shows the boxplots for both maximum and minimum joint angles and the results of the post-hoc pairwise tests regarding AFO variability. From Figure 17, we can observe that knee joint angle values are generally lower than the control situation, while hip angles have higher values when wearing an AFO.

Table 9 and Table 10 present the results for maximum and minimum joint angles for the conditions where the ANOVA *p*-value was <0.01. The mean and standard deviation are also presented.

We found statistically significant differences between conditions in six of the twelve joint angles considered. These were the maximum angle values of the right and left knees and the left hip, as well as the minimum angle values of the right ankle and both hips. Of these six results, in all of them, the H2-AFO was significantly different from the control condition, while the F-AFO and L-AFO were significantly different from the control only for the minimum angles of the hip joints. The F-AFO and L-AFO were significantly different from the H2-AFO only for the maximum angles of the left hip. No differences were found between the F-AFO and the L-AFO.

Regarding the spatial-temporal parameters, the mean and standard deviation values, as well as ANOVA and Tukey’s HSD *p*-values (if <0.05) for each parameter at self-selected and controlled speeds, can be found in Table 11 and Table 12, respectively. Only parameters for the right leg (with AFO) are presented since the results are largely similar for the left leg. Full results for both limbs can be found in Appendix D.

The statistical tests showed significant differences in all spatiotemporal parameters between trials of self-selected and controlled speeds. When the users are walking at a self-selected speed, pairwise post-hoc tests revealed differences between the H2-AFO and the other three conditions (team-developed AFOs and control) for all parameters. Further, the gait with F-AFO and L-AFO at self-selected speed presents similar spatiotemporal parameters when compared with the control condition, except for the swing time duration of the right foot (*p*-values of 0.0038 and 0.0025, respectively).

Different results were observed when walking at controlled speeds. First, post-hoc tests show no difference between the H2-AFO and the other two AFOs regarding most spatiotemporal parameters, except for the left stance time. The H2-AFO is still significantly different from the control condition for all parameters, while the F-AFO and L-AFO only present statistically significant differences regarding temporal parameters. When statistical differences were found, temporal parameters had larger values for the AFOs when compared to the control gait, while spatial parameters had smaller values.

Finally, we verified that the use of an AFO creates a significant gait asymmetry for stance and swing times (*p*-values < 0.0001 when compared with the control condition). In fact, wearing an AFO leads to the right leg spending less time in the stance phase than the left leg while spending more time in the swing phase.

## 5. Discussion

Literature shows that human-robot joint misalignment leads to spurious forces and torques at HRi [15,16,18,19,46]. These forces can be caused by pressure interactions (forces normal to the tissue surface) above safe levels or by sheer/friction interactions (forces tangent to the tissue surface) [13]. As such, this study aims to directly assess misalignment through motion capture data [21,23,47,48] and directly or indirectly assess these interactions. To achieve this goal, pressure sensors were placed at the HRi that allow direct quantification of the pressure interactions [26,27,28,49,50], while the motion capture data allows the capture of relative movement between the AFO’s fixations (e.g., cuffs, straps) and the human limb, giving an indirect assessment of sheer interactions. Higher recorded misalignments are expected to lead to higher recorded interactions. Finally, this study issued a holistic assessment by combining objective measures of misalignment with user-perceived comfort and satisfaction. This perception was assessed through a modified version of the CSD-OPUS questionnaire [40].

Motion data analysis allowed direct conclusions regarding the human-AFO joint misalignment and relative displacement between the fixations and the human limb. *Linear misalignment measures* (Figure 9) show differences in the vertical axis, although the F-AFO and L-AFO have a lower initial misalignment than the H2-AFO. No significant differences were observed for the initial misalignment in the horizontal axis, which goes against our initial hypothesis that there were significant differences among all AFOs. Furthermore, misalignment varies during gait. This may be caused either by slippage of the connections or migration of the instantaneous center of rotation (ICR) of the biological joint, which, due to the kinematic mismatch between both joints, is not followed by the AFO [51]. This ICR migration has been reported as one of the main causes of human-robot joint misalignment during gait [15,17,19]. The vertical misalignment diminishes in the H2-AFO during the swing phase of the gait. However, it is important to note that, during this gait phase, theoretically, power transmission from the robotic joint to the user is not needed. In fact, it is to be expected that the largest torques are present during stance, and, as such, misalignment during this phase will lead to higher spurious forces and torques at the HRi than during the swing phase. These considerations are supported by literature, where it was found that force and torque interactions reach their peak during late stance [19,26]. As such, joint misalignment during stance represents a more relevant safety hazard. The F-AFO and L-AFO have statistically significant lower vertical misalignments in both the stance and swing phases (*p*-values of 0.0022 and 0.0017 for stance and <0.0001 and 0.0015 for swing) than H2-AFO, indicating lower misalignment-related interactions at the HRi for team-developed AFOs. Initial misalignments for our study are approximately 1.00 cm (horizontal) and 1.50 cm (vertical). These values are within the magnitude of values found in [21,22] (−5.50 cm and 1.40 cm vertical and −3.00 cm and 2.30 cm horizontal, respectively). The lower misalignment values achieved in our study are due to the smaller range of motion of the ankle joint in the sagittal plane in comparison to the knee joint assessed in [44,45].

Regarding angular misalignment measures (Figure 10), we find differences in the curves of the custom-built AFOs in comparison with the commercial AFO for the first 25% of gait. While this can be attributed to the effect of the actuation module of the H2 device that pushed against the user, these differences were not statistically significant. As such, this measure does not provide significant answers to our research questions.

Displacement along the Z axis is the value that can more closely be correlated with sheer or frictional interactions, as described in [11]. The results show higher displacements along the Z axis for H2-AFO during the initial stance phase and for F-AFO and L-AFO during the swing phase. Nevertheless, the displacements present a small magnitude, and the differences in the maximum displacements between H2-AFO, F-AFO, and L-AFO are within 2 mm. This value is in the same order of magnitude as the error reported in [23] for fixation displacement and, as such, cannot be separated from some systematic or occasional error that could have occurred during the protocol. Therefore, all three AFOs lead, approximately, to the same sheer or friction interactions. The work done by [23] showed maximum vertical displacements of 0.4 cm, while in [24], maximum *vertical displacements* of 2.5 cm were reported. Our work reports average *vertical displacements* between 0.19 and 0.33 cm, which is within the magnitude of the values found in the literature. The study [24] reported that high values of *vertical displacements* are mainly associated with cuff slippage, while low displacements may be related to soft-tissue deformation. The results from our work are below the values reported by [24], which may indicate that the displacements observed in the three AFOs may be associated with soft-tissue deformation.

Regarding *displacement angles*, significant differences between AFOs were only observed around the sagittal plane (*Dα* measure in Figure 12, *p*-values < 0.0001). This result may be explained by the quasi-passive assistance provided by the H2-AFO in the sagittal plane, while the other AFOs provided passive assistance in all planes. Nevertheless, the *displacement angles* obtained for the three AFOs have approximately the same values and shape throughout the gait cycle. Thus, this measurement provides no significant answers to our research questions.

Pressure data analysis shows fewer pressure interactions for the frontal design (F-AFO) than the lateral design (L-AFO) and H2-AFO. Safety assessment, however, is also fundamental to ascertaining if the measured pressures are below safety values [13,17], as presented in Table 8. Results indicate that the peak pressure values are largely below PPT values from single-point algometry for all three AFOs. These results are an initial benchmark to assess safety and pain onset.

However, the pressures related to AFO fixations are usually applied throughout a large area and not at single discrete points. Another difference between single-point algometry and CPA is that the latter stimulates deep tissues (e.g., deep muscle layers over bone). As such, ensuring HRi pressure below CPA threshold values can prevent the occurrence of both superficial and deep tissue injuries [43]. Since these thresholds are primarily valid for pressures measured at the calf, sensors located posteriorly were used [52]. In comparing the pressures recorded by the sensors, we assumed that they may represent the pressure distribution of the HRi. This assumption holds true for all sensors in both groups because they were all on the same plane as the straps. It is important to note that CPA pressures depend on cuff width and cannot be directly equated to interface pressures measured by our sensors [52]. Despite this, we used CPA thresholds as reference points to distinguish between orthoses. Higher FSR pressure values than PDT thresholds on interfaces do not necessarily imply discomfort for most users. However, when combined with other data like misalignment measures and user-perceived comfort, it raises concerns about the safety of that specific interface. Comparing benchmarks from single-point algometry alone does not support this comprehensive approach and does not yield conclusive results for assessing the safety of a cuff-based HRi.

The relatively high peak pressure values for the L-AFO in comparison to the F-AFO may be attributed to the type and material used in the straps since these AFOs report similar misalignments [12,13]. Further, the value for the anterior proximal group of the H2-AFO surpasses safety levels even in healthy subjects, which is a major counter-indication for its use. Furthermore, the posterior group shows values close to detection thresholds, suggesting a contraindication for using the posterior fixation mechanism embedded in H2-AFO.

The peak pressures recorded for the H2-AFO are higher than both F-AFO and L-AFO, which may be explained, in part, by the human-joint misalignment. Overall, recorded values for the F-AFO are largely within safety levels from the literature [17,42,43], supporting its adoption. This study did not find significant changes in HRi between the data recorded for the participants wearing skin-tight long tights in comparison with the participants wearing sports shorts.

Questionnaire results show higher comfort scores for the F-AFO. When comparing F-AFO and H2-AFO, the F-AFO’s shank design (i.e., frontal shin guard) and foot design (sports shoes with a flexible outsole) presented higher user satisfaction than the H2-AFO’s lateral straps and rigid outsole. The main differences between F-AFO and L-AFO lie in the fixations at the level of the shin guard. While the frontal model relies on Velcro fixations, the lateral model relies on straps made of a stiffer polymer and a tighter fit, resulting in pain, discomfort, abrasions, and irritations, in accordance with the higher-pressure values recorded. These effects can be explained by the fact that a strap-based system (L-AFO) may allow for higher fitting pressure (the initial pressure at the interface after fitting the device to the user) than a Velcro-based system (F-AFO). Furthermore, by having the shin guard located laterally, the anterior straps are in contact with the bony prominences of the tibia (an anatomical area with lower pressure thresholds for pain [13,17]). This contrasts with the F-AFO design, where the location of the shin guard ensures a better distribution of pressure, reducing pain and discomfort. Two studies [53,54] have used the NASA TLX questionnaire [55], which assesses task workload on the user, and a simple 0–100 analog scale for assessing the user’s comfort. However, when these studies found significant comfort scores between test conditions (e.g., AFOs with and without solutions to increase comfort), no significant difference was found in NASA TLX scores. On the other hand, the questionnaire used in this work found significant differences (*p*-values of 0.016 and 0.006) between the three AFOs regarding user satisfaction and allowed us to assess which AFO’s characteristics most contributed to these differences. The same has not been observed in the literature. Further, within the scope of the research done, this is the first work that verified a correlation and concordance between the user-reported outcomes and quantitative measures of human-robot misalignment.

In addition to comparing the three fixation designs, this study also analyzed the AFOs’ effects on gait kinematics. Regarding joint angles in the sagittal plane, some conclusions can be drawn. First, differences in ankle angles were only observed for the right ankle between the control condition and the H2-AFO. This shows that the H2-AFO restricts the normal ROM of the plantarflexion-dorsiflexion motion, which is a counter-indication for its use. There are also differences in the level of the knee and hip joints. Only the H2-AFO significantly reduced the ROM of both knee joints in comparison with the control (*p*-values < 0.01), which does not favor its use, while no significant differences were found between the F-AFO/L-AFO and the No-AFO condition. Furthermore, results reported increased angles of the hip joint for all three AFOs in comparison with the control condition. It may result from a compensatory motion due to the increased inertia associated with the presence of a distal mass. These results agree with the literature, namely studies [44,45], where higher hip joint angles and lower knee joint angles were observed when wearing the Lokomat exoskeleton when compared to the control gait. Overall, all three AFOs changed the gait patterns at the hip joint level, and the H2-AFO also interfered with the knee and ankle joints. This can be largely attributed to the fixation mechanism since there were no speed-related differences.

Finally, by analyzing the spatiotemporal parameters, we observed the following aspects: The self-selected gait speed cannot be maintained when using the H2-AFO, while this is possible for the F-AFO and L-AFO. In this condition, the F-AFO and L-AFO show no significant deviation regarding the natural gait patterns for all parameters except for the right swing time. This is mainly due to the inertial effect of the mass of the AFO, which is greater during the first half of the swing phase (towards its maximum foot clearance) because of gravity and will increase the time spent in this phase of the cycle. Results also show that this effect is not compensated during the second half of the swing phase, which can indicate that this inertial effect is felt throughout the entirety of the swing phase. When all AFOs were used at the controlled speed, they caused similar effects in temporal parameters, with the only difference being in step length, where the user still maintained normal values when wearing the F-AFO and L-AFO.

Overall, the F-AFO presents lower initial misalignments and pressures, higher user-perceived comfort, and less disturbance of normal gait patterns in comparison with the H2-AFO and L-AFO. These findings allow us to answer the research question of this work: that an AFO with a frontal shin guard and sports shoe may present a more appropriate HRi. Overall, our hypothesis that there were significant differences among AFOs regarding misalignment and gait kinematics was only proven when comparing the H2-AFO with the F-AFO and L-AFO. In fact, we only found significant differences between the two orthoses of the SmartOs project when assessing user-perceived comfort.

### Study Limitations

The main limitation of our work is that the H2-AFO includes an actuator, while the F-AFO and L-AFO do not. This limitation was tackled both (i) by setting the passive mode of the H2-AFO’s actuator and (ii) by fixing a structure at the ankle joint level of the F-AFO and L-AFO that mimicked the body mass and shape of their actuation systems. Further experiments should be carried out using the three AFOs with the actuation system.

Another limitation is that the pressure values were measured by FSRs in discrete locations in the HRi. These results are sensitive to errors in the FSRs’ placement and do not enable the measurement of pressure distributions throughout the shank. While an effort was made to place the FSRs in the same anatomical landmarks across subjects and AFOs, an assessment of pressure distribution around the entire HRi should be done in a future study.

The last limitation is that the involved participants are not the target end-users of these AFOs (i.e., stroke patients). Since both age and neurological conditions are factors that lower the pain and comfort threshold for pressure [42], the questionnaire results might change for stroke patients.

## 6. Conclusions

In this work, we designed and carried out an experimental protocol to determine which of three AFOs (a commercially available exoskeleton H2-AFO and two prototypes developed in-house with a frontal, F-AFO, or lateral, L-AFO, shin guards) presents lower misalignment, interactions at the HRi, changes in gait patterns, and higher user-perceived comfort.

Findings show that both F-AFO and L-AFO improve on the H2-AFO design, presenting a reduced misalignment at the beginning and during gait. Consequently, both F-AFO and L-AFO presented reduced pressure values and higher user-perceived comfort and allowed for a gait pattern that is closer to a healthy gait. Within the F-AFO and L-AFO, the frontal shin guard leads to reduced pressure and higher user-perceived comfort. In conclusion, this study pointed out that an AFO with a frontal shin guard and a foot modulus based on a sports shoe presents reduced misalignment and more appropriate physical HRi regarding a lateral shin guard and an outsole. This provides a good indication for the adoption of this design in future studies.

## Figures and Tables

**Figure 1 sensors-24-00246-f001:**
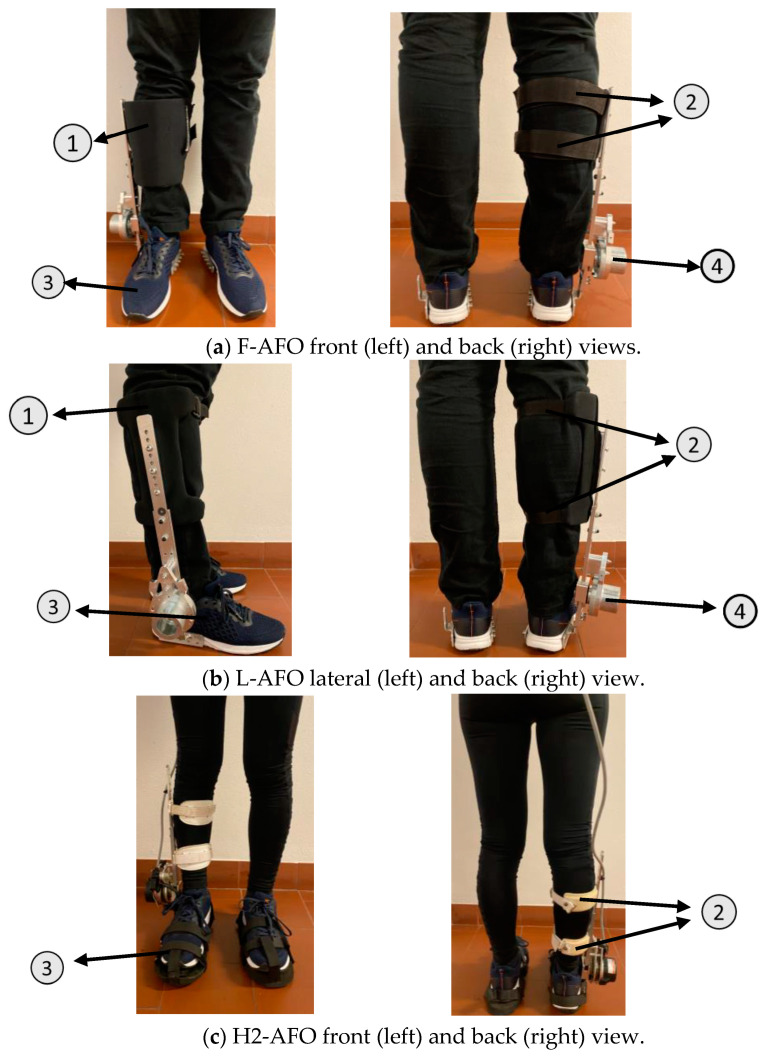
Representation of the modules of the three AFOs used in this study. (**a**) F-AFO (team-developed AFO). (**b**) L-AFO (team-developed AFO). (**c**) H2-AFO (Technaid S.L.). 1—Shin-guards. 2—Shank fixation (straps/cuffs). 3—Foot fixation. 4—Actuation structure.

**Figure 2 sensors-24-00246-f002:**
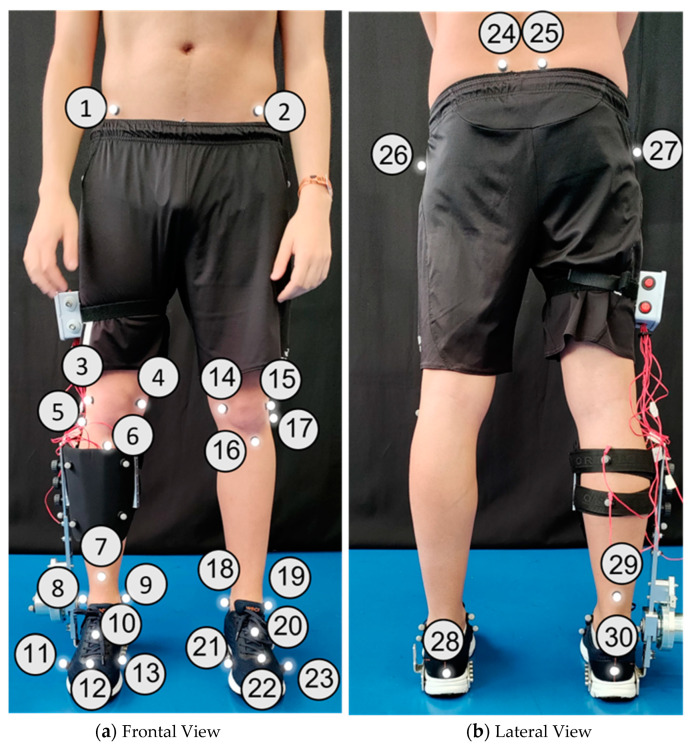
Marker set for the human body modeling. Numbers identify each marker used for the biomechanical model.

**Figure 3 sensors-24-00246-f003:**
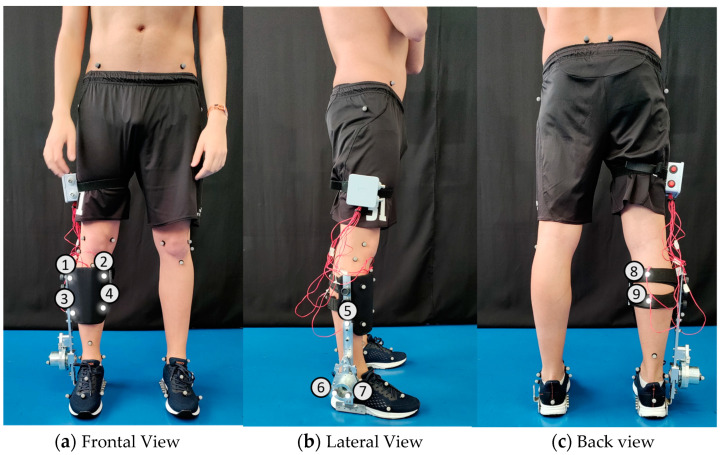
Marker set for the F-AFO. Numbers identify each marker used for the mechanical model.

**Figure 4 sensors-24-00246-f004:**
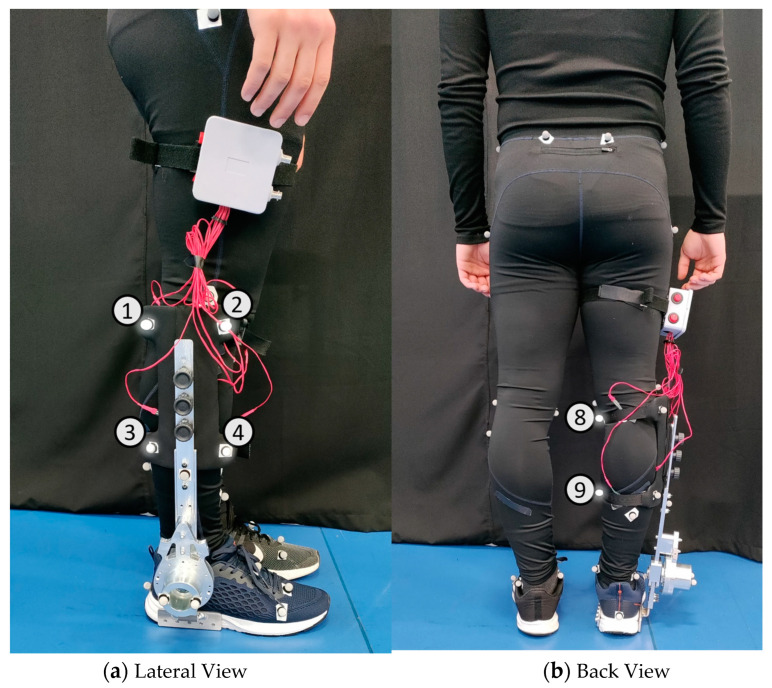
Marker set for the L-AFO. Numbers identify each marker used for the mechanical model.

**Figure 5 sensors-24-00246-f005:**
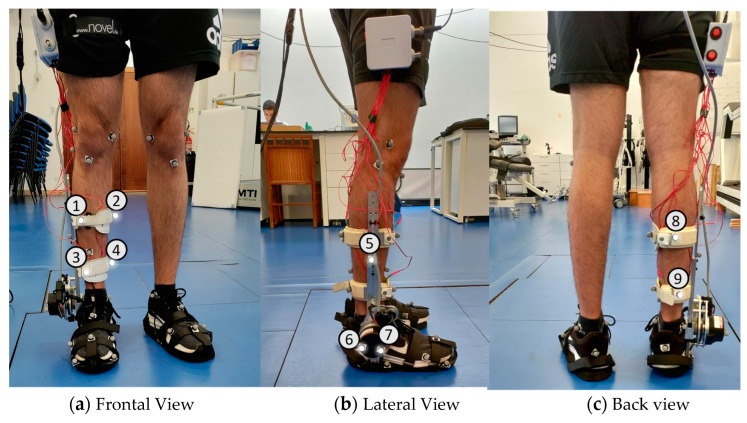
Marker set for the H2-AFO. Numbers identify each marker used for the mechanical model.

**Figure 6 sensors-24-00246-f006:**
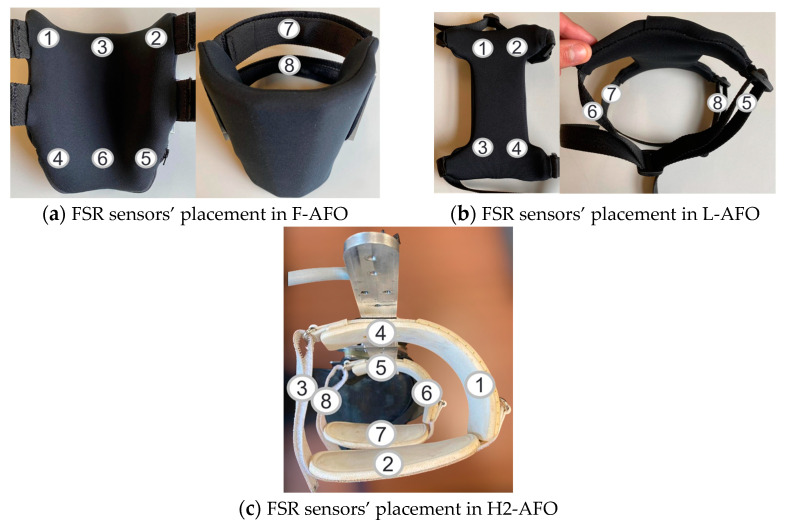
FSR location for each AFO. Numbers identify each sensor used for capturing interface pressure.

**Figure 7 sensors-24-00246-f007:**
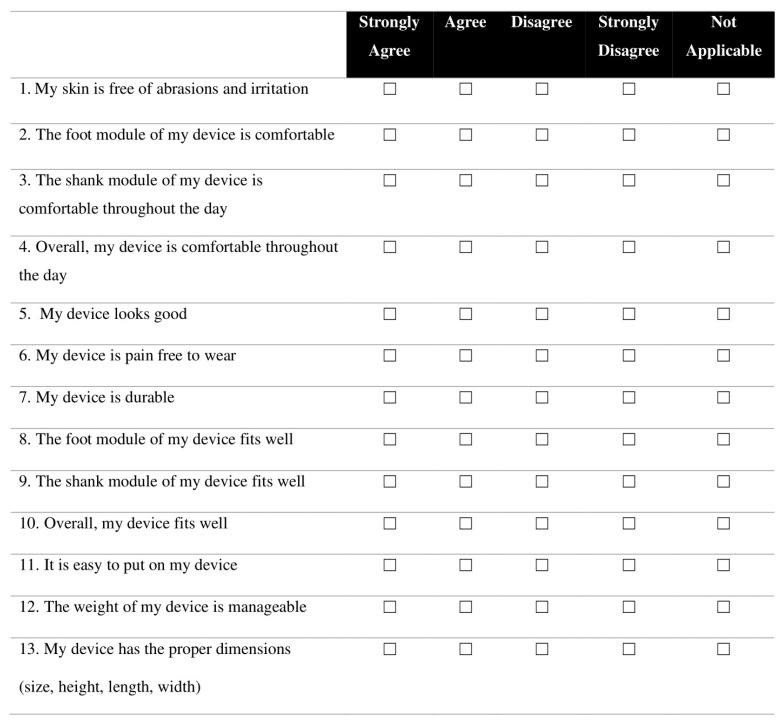
Questionnaire given to the participants, based on [40].

**Figure 8 sensors-24-00246-f008:**
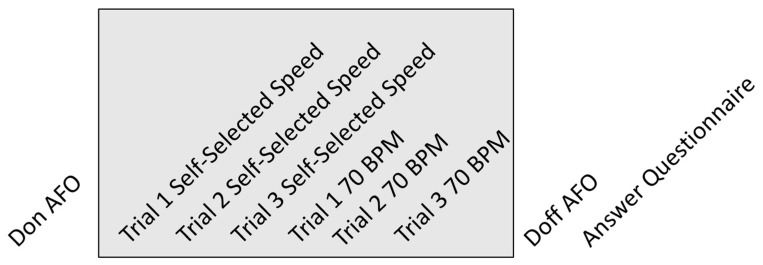
Schematic of the protocol, repeated for each AFO in a randomized sequence.

**Figure 9 sensors-24-00246-f009:**
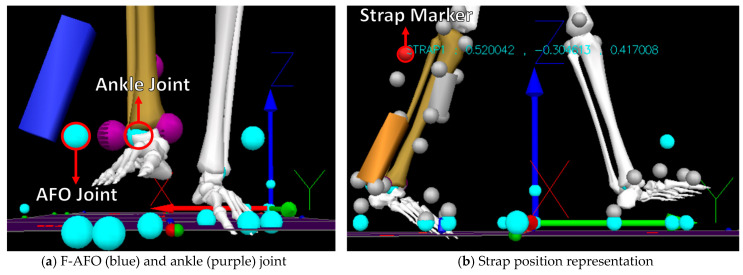
AFO and Human models for calculating misalignment and displacement.

**Figure 10 sensors-24-00246-f010:**
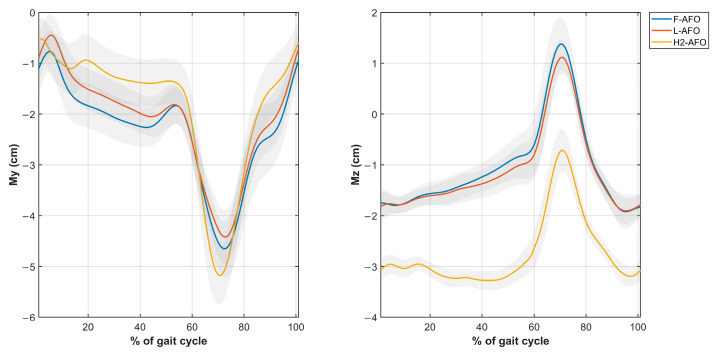
Average and standard deviation of horizontal (*My*, **left** view) and vertical (*My*, **right** view) misalignment.

**Figure 11 sensors-24-00246-f011:**
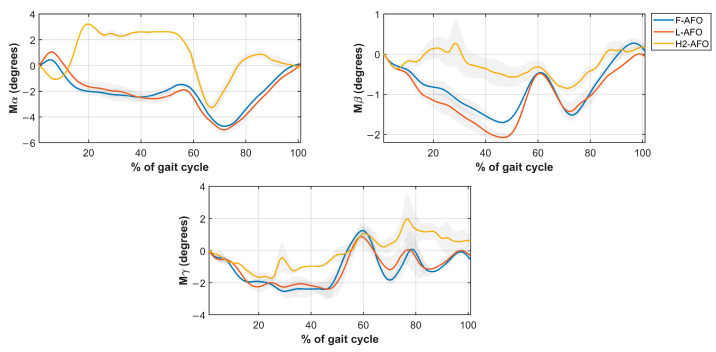
Average and standard deviation of the *Misalignment angle Mα* (**top left**), *Mβ* (**top right**), and *Mγ* (**bottom middle**).

**Figure 12 sensors-24-00246-f012:**
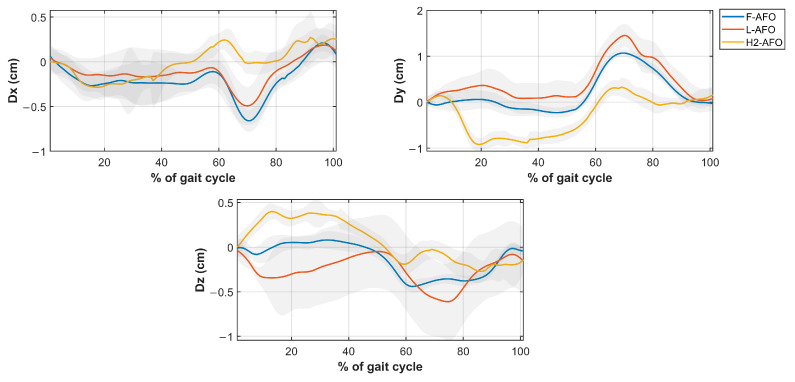
Average and standard deviation of the *linear displacement* along the X (**top left**), Y (**top right**), and Z (**bottom middle**) axis.

**Figure 13 sensors-24-00246-f013:**
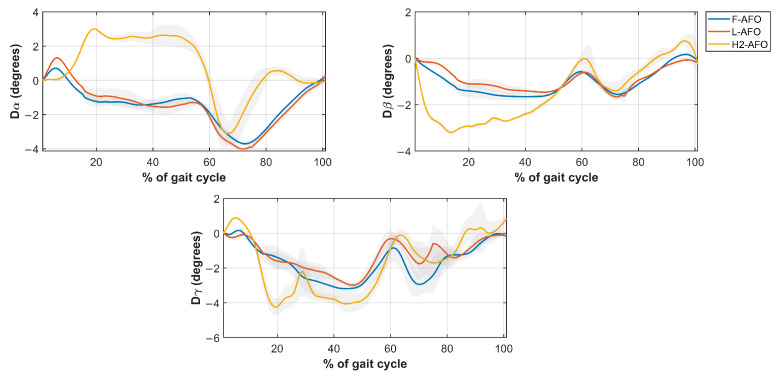
Average and standard deviation of the *angular displacement Dα* (**top left**), *Dβ* (**top right**), and *Dγ* (**bottom middle**).

**Figure 14 sensors-24-00246-f014:**
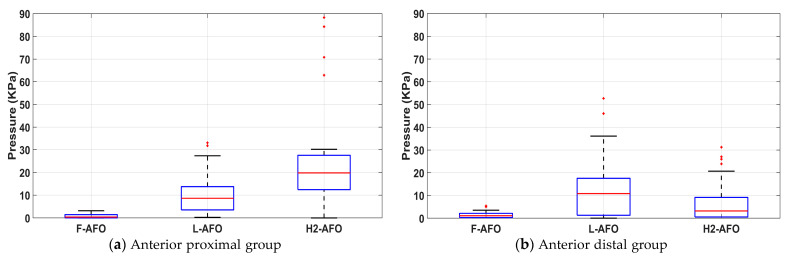
Boxplots of peak pressure values.

**Figure 15 sensors-24-00246-f015:**
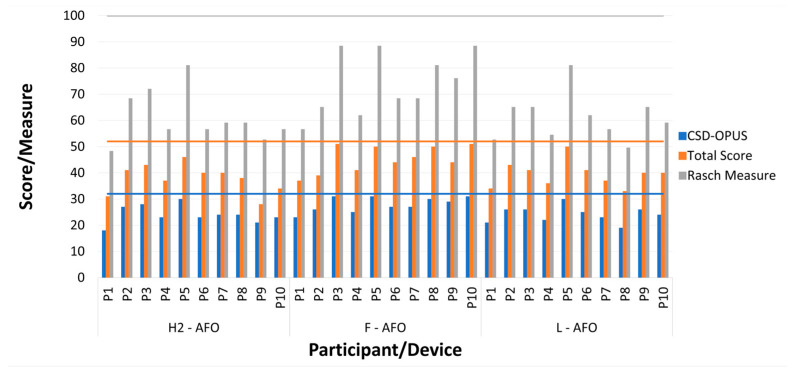
Questionnaire scores were presented for each participant (P1–P10) and AFO. Blue bars correspond to the score of the questions from CSD-OPUS, orange bars to the total score of the questionnaire, and grey bars to the corresponding Rash Measures. The maximum values for each score are 32, 52, and 100, respectively, represented by horizontal bars of the same color.

**Figure 16 sensors-24-00246-f016:**
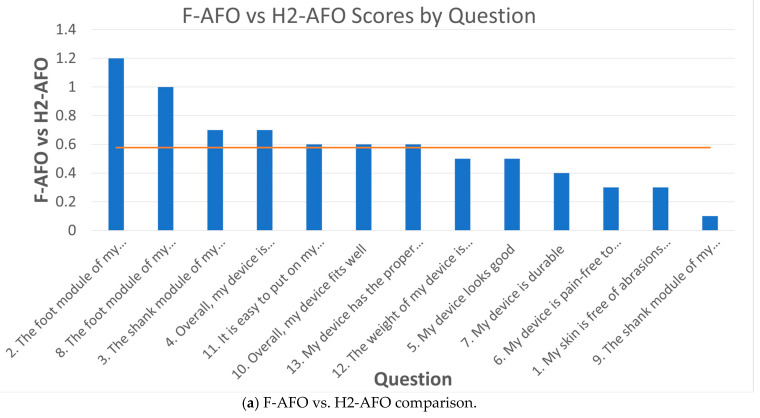
Comparison of rated scores for each question. Orange lines represent the average difference across all questions.

**Figure 17 sensors-24-00246-f017:**
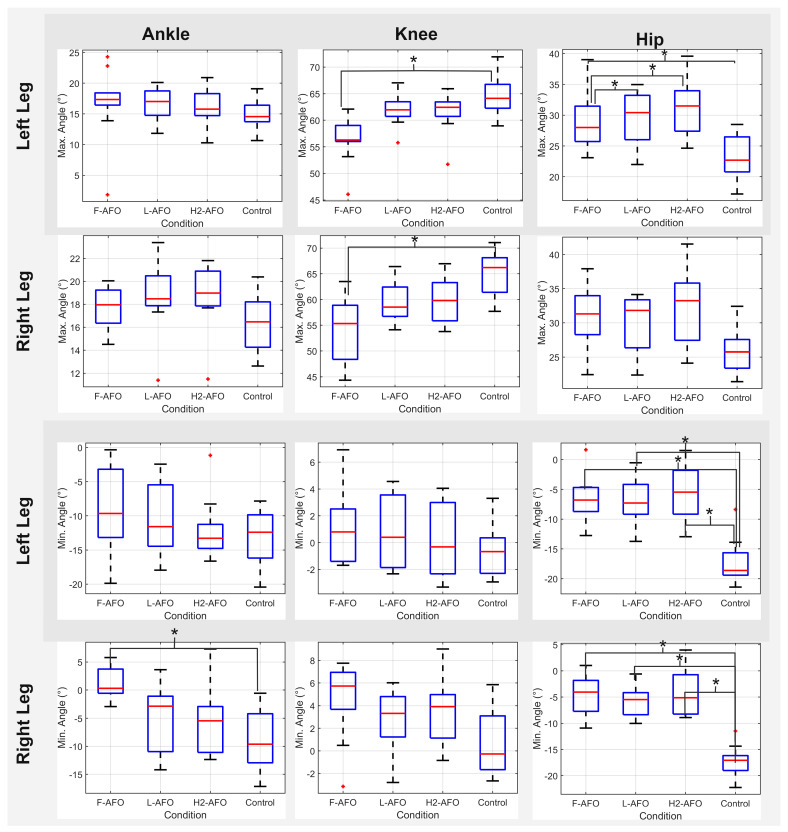
Maximum and minimum values for joint angles for right and left ankle, knee, and hip joint. Asterisks represent conditions where the post-hoc Tuskey HSD test resulted in a * *p*-value < 0.01.

**Table 1 sensors-24-00246-t001:** Characteristics of AFOs used in this study.

	H2-AFO	SOF-AFO	SOL-AFO
Foot Module	Rubber Outsole with 4 Velcro straps	Sports shoe	Sports shoe
Shin Guard	NA	Frontal	Lateral
Shank Fixation	Lateral and posterior foam Velcro straps	Posterior Velcro straps	Lateral and posterior fabric straps
Upright structure	Stainless steel, 7005 aluminum	AL5754 aluminum	AL5754 aluminum
Mass (kg)	2.10	2.18	1.90

**Table 2 sensors-24-00246-t002:** Average strap distance in cm and as a percentage of shank length (in parentheses).

	H2-AFO	SOF-AFO	SOL-AFO
Top Strap	19.56 ± 3.07 (0.48)	8.66 ± 2.84 (0.21)	7.3 ± 1.43 (0.18)
Bottom Strap	32.96 ± 2.86 (0.80)	17.35 ± 4.86 (0.42)	24.56 ± 2.45 (0.60)

**Table 3 sensors-24-00246-t003:** FSR grouping per anatomical cluster. The number corresponds to the identification of FSR indicated in Figure 6.

FSR Group	F-AFO	L-AFO	H2-AFO
Anterior Proximal	3	6	3
Anterior Distal	6	7	8
Posterior Proximal	7	5	1
Posterior Distal	8	8	6

**Table 4 sensors-24-00246-t004:** Statistical tests carried out in this work. All null hypotheses were tested for a 5% significance level except for the ones comparing joint angles, which were tested for a 1% significance level. All measures were averaged for all subjects for a given studied condition.

Null Hypothesis	Studied Condition	Fixed Conditions	Measure	Test
No differences between gait speeds	Gait Speed Variability	FSR, Orthosis	Average Linear and Angular Misalignment/Displacement during Stance and Swing	Two-sample *t*-test
No differences between the 3 AFOs	Orthosis Variability	Speed, Trial, Measure	Average Linear and Angular Misalignment/Displacement during Stance and Swing	One-Way ANOVA/Tukey’s HSD
No differences between gait speeds	Gait Speed Variability	FSR, Orthosis	Peak Pressure	Wilcoxon signed-rank Test
No differences in Rash Measures between 3 AFOs	Orthosis Variability	FSR, Speed, Trial	Rash Measures	Friedman Test/Tukey’s HSD
Joint angles have equal means and variances at different gait speeds	Speed Variability	Orthosis, Joint	Maximum/Minimum Joint Angle	Two-sample *t*-test
No difference in maximum/minimum joint angles between the 4 conditions	Orthosis Variability	Speed, Trial, Joint	Maximum/Minimum Joint Angle	One-Way ANOVA/Tukey’s HSD
No difference in Spatiotemporal Parameters between 3 AFOs	Orthosis Variability	Speed, Trial, Parameter	Spatiotemporal Parameters	One-Way ANOVA/Tukey’s HSD

**Table 5 sensors-24-00246-t005:** Mean, standard deviation, ANOVA, and Tukey’s HSD (when <0.05) post hoc *p*-values of computed measures during the stance phase of gait. Each symbol represents a pairwise comparison between the condition of a given column and one of the other conditions. * Comparison with F-AFO. † Comparison with L-AFO.

Measure	H2	SOF	SOL	ANOVA *p*-Value
Mz (cm)	−3.03 ± 0.24	−1.38 ± 0.31	−1.33 ± 0.11	0.0006
	* 0.0022 † 0.0017			
Dy (cm)	−0.36 ± 0.19	0.2 ± −0.69	0.44 ± −0.35	<0.0001
	* <0.05 † <0.05	† 0.0231		
Dz (cm)	0.19 ± 0.13	−0.18 ± −0.91	−0.15 ± −1.2	0.0036
	* 0.0070 † 0.0112			
Dα (°)	1.51 ± 0.66	−1.05 ± 0.45	−0.86 ± 1.06	<0.0001
	* <0.0001 † <0.0001			

**Table 6 sensors-24-00246-t006:** Mean, standard deviation, ANOVA, and Tukey’s HSD (when <0.05) post hoc *p*-values of computed measures during the swing phase of gait. Each symbol represents a pairwise comparison between the condition of a given column and one of the other conditions. * Comparison with F-AFO. † Comparison with L-AFO.

Measure	H2	SOF	SOL	*p*-Value
Mz (cm)	−2.15 ± 0.31	−0.48 ± 0.38	−0.58 ± 0.38	0.0003
	* <0.0001 † 0.0015			
Dx (cm)	−0.2 ± 0.7	−0.45 ± 0.33	−0.32 ± −0.87	0.0170
	* 0.0126			
Dy (cm)	0.14 ± −0.56	0.88 ± −0.83	1.08 ± −0.38	<0.0001
	* <0.0001 † <0.0001			
Dz (cm)	−0.1 ± −2.01	−0.3 ± −0.69	−0.33 ± −0.65	0.0103
	* 0.0309 † 0.0152			
Dα (°)	−0.55 ± 1.13	−2.3 ± 0.32	−2.54 ± 0.88	0.0020
	* <0.001 † <0.0001			

**Table 7 sensors-24-00246-t007:** The *p*-values obtained from Wilcoxon signed ranked tests to test the null hypothesis that there are no statistically significant differences in the peak pressure values between the two different speeds for three AFOs.

	H2-AFO	F-AFO	L-AFO
FSR1	1.000	1.000	1.000
FSR2	1.000	0.375	1.000
FSR3	1.000	0.508	1.000
FSR4	0.727	1.000	1.000
FSR5	0.453	0.727	0.508
FSR6	0.727	0.039	0.039
FSR7	1.000	1.000	1.000
FSR8	0.727	0.508	1.000

**Table 8 sensors-24-00246-t008:** Results of measured peak pressure values and literature values for PDT, PTT, and single-point PPT [17,42,43].

	Pressure Values (kPa)							
	F-AFO	L-AFO	H2-AFO	PDT		PTT		PPT
				Healthy	Chronic Pain	Healthy	Chronic Pain	
Group Anterior Proximal	0.88 ± 1.01	10.84 ± 8.8	24.66 ± 20.72	NA				545.50
Group Anterior Distal	1.43 ± 1.26	12.55 ± 12.88	6.91 ± 8.32					588.10
Group Posterior Proximal	7.52 ± 8.46	8.98 ± 6.23	9.42 ± 7.44	16–34	10–18	42–91	<25	416.60
Group Posterior Distal	2.89 *±* 1.73	12.02 *±* 7.7	11.68 *±* 7.63					416.60

**Table 9 sensors-24-00246-t009:** Maximum joint angles (Mean ± standard deviation) and ANOVA *p*-values (if <0.01) for comparing AFO and non-AFO conditions.

Condition	Right Knee	Left Knee	Left Hip
H2-AFO	54.24 ± 6.55°	56.34 ± 4.52°	29.14 ± 5.17°
F-AFO	59.65 ± 4.12°	61.82 ± 2.95°	29.71 ± 4.34°
L-AFO	59.72 ± 4.71°	61.59 ± 3.92°	31.27 ± 4.68°
Non-AFO (Control)	65.36 ± 4.32°	64.41 ± 3.78°	23.17 ± 3.67°
*p*-Value	0.00038	0.00039	0.00333

**Table 10 sensors-24-00246-t010:** Minimum joint angles (Mean ± standard deviation) and ANOVA *p*-values (if <0.01) for comparing AFO and non-AFO conditions.

Orthosis	Right Ankle	Right Hip	Left Hip
H2-AFO	1.32 ± 2.84°	−4.59 ± 4.1°	−6.54 ± 4.19°
F-AFO	−5.34 ± 6.67°	−5.74 ± 2.94°	−7.13 ± 4.23°
L-AFO	−5.6 ± 6.31°	−4.5 ± 4.48°	−5.78 ± 4.86°
Non-AFO (Control)	−8.5 ± 5.54°	−17.28 ± 3.06°	−17.24 ± 3.97°
*p*-Value	0.0065	<0.0001	<0.0001

**Table 11 sensors-24-00246-t011:** Mean, standard deviation, ANOVA, and Tukey’s HSD (if <0.05) post hoc *p*-values of computed spatiotemporal parameters for self-selected speed. Each symbol represents a pairwise comparison between the condition of a given column and one of the other conditions. L/R represents a ratio between the left and right limbs for that parameter. * Comparison with F-AFO. † Comparison with L-AFO. ‡ Comparison with Control.

Parameter	H2-AFO	F-AFO	L-AFO	Control	*p*-Value
Gait Speed	0.64 ± 0.12	0.93 ± 0.09	0.94 ± 0.13	1.07 ± 0.09	<0.0001
	* <0.0001 † <0.0001 ‡ <0.0001				
Right Step Length	0.57 ± 0.04	0.63 ± 0.04	0.64 ± 0.05	0.64 ± 0.03	0.0021
	* 0.0197 † 0.0001 ‡ 0.0001				
Right Stance Time	1.18 ± 0.22	0.84 ± 0.08	0.85 ± 0.1	0.77 ± 0.06	<0.0001
	* <0.0001 † <0.0001 ‡ <0.0001				
Right Swing time	0.67 ± 0.08	0.53 ± 0.05	0.53 ± 0.05	0.43 ± 0.03	<0.0001
	* <0.0001 † <0.0001 ‡ <0.0001	‡ 0.0038	‡ 0.0025		
L/R Stance Time	1.1 ± 0.04	1.05 ± 0.03	1.05 ± 0.03	1 ± 0.02	<0.0001
	* 0.0094 † 0.0093 ‡ <0.0001	‡ <0.0001	‡ <0.0001		
L/R Swing Time	0.83 ± 0.05	0.91 ± 0.04	0.91 ± 0.04	1.01 ± 0.04	<0.0001
	* 0.0011 † <0.0001 ‡ <0.0001	‡ <0. 0001	‡ <0.0001		

**Table 12 sensors-24-00246-t012:** Mean, standard deviation, ANOVA, and Tukey’s HSD (if <0.05) post hoc *p*-values of computed spatiotemporal parameters for the controlled speed. Each symbol represents a pairwise comparison between the condition of a given column and one of the other conditions. L/R represent a ratio between the left and right limbs for that parameter. ‡ Comparison with Control.

Parameter	H2-AFO	F-AFO	L-AFO	Control	*p*-Value
Gait Speed	0.67 ± 0.05	0.73 ± 0.06	0.73 ± 0.07	1.07 ± 0.09	<0.0001
	‡ <0.0001	‡ <0.05	‡ <0.0001		
Right Step Length	0.58 ± 0.04	0.62 ± 0.04	0.62 ± 0.05	0.64 ± 0.03	0.0087
	‡ 0.0067				
Right Stance Time	1.11 ± 0.07	1.08 ± 0.05	1.07 ± 0.05	0.77 ± 0.06	<0.0001
	‡ <0.0001	‡ <0.0001	‡ <0.0001		
Right Swing time	0.64 ± 0.05	0.64 ± 0.03	0.65 ± 0.04	0.43 ± 0.03	<0.0001
	‡ <0.0001	‡ <0.0001	‡ <0.0001		
L/R Stance time	1.08 ± 0.06	1.03 ± 0.02	1.04 ± 0.04	1 ± 0.02	0.00036
	‡ <0.0001				
L/R Swing Time	0.88 ± 0.11	0.95 ± 0.06	0.92 ± 0.05	1.01 ± 0.04	0.00294
	‡ 0.0016				

## Data Availability

Data are contained within the article and Appendix A.

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
