# Peer review of "Human-Robot Joint Misalignment, Physical Interaction, and Gait Kinematic Assessment in Ankle-Foot Orthoses"

_sensors, 2023, doi:10.3390/s24010246_

Round 1

Reviewer 1 Report

Comments and Suggestions for Authors

I would like to thank the authors for their clarifications and manuscript revision. My concerns have been sufficiently addressed, and the manuscript will be publishable after a minor revision.

Comments on the Quality of English Language

Please, proofread the manuscript thoroughly; some issues are listed below:

"The protocol ended by donning off the AFO ..." -> "The protocol ended by doffing the AFO ..."

"As such, ensuring HRi pressure below CPA threshold values CPA can prevent the occurrence of both superficial and deep tissue injuries [42]." - Is the second "CPA" in this sentence superfluous?

"... and cannot be directly equated to interface pressure measured by our sensors." -> "... and cannot be directly equated to interface pressures measured by our sensors."

"... can be measured trough single point algometry ..." -> "... can be measured through single point algometry ..."

Author Response

We thank the reviewer for the comments and suggestions provided. Please see the attachment for our response.

Reviewer 2 Report

Comments and Suggestions for Authors

The authors have addressed most of the concerns raised by the reviewer. One small suggestion is to redraw Figure 8 with a better representation and revise the caption for a better understanding of the protocol used.

Author Response

(The authors gave the same response as above.)
